# Modelling the associations between faculty support and academic satisfaction among nursing students: Mediating roles of sleep problems and smartphone addiction

Jerry Paul K. Ninnoni[1], Ignatius N. Ijere[2], Isaac Tetteh Commey[1]*, Mustapha Amoadu[3]

1 Department of Mental Health, School of Nursing and Midwifery, University of Cape Coast, Cape Coast, Ghana, 2 Department of Public Health, Syracuse University, New York, United States of America, 3 Biomedical and Clinical Research Centre, University of Cape Coast, Cape Coast, Ghana

* isaac.commey@ucc.edu.gh

## Abstract

Faculty support plays a vital role in shaping academic satisfaction among university students. However, its indirect effects through behavioural and psychological pathways such as sleep problems and smartphone addiction, have received limited research attention, particularly in resource-limited contexts like Ghana. This study modelled the association between perceived faculty support and academic satisfaction among student nurses, with sleep problems and smartphone addiction examined as mediating variables. A cross-sectional survey was conducted among 563 nursing students at the University of Cape Coast, Ghana. Standardised tools were used to assess perceived faculty support, sleep problems, smartphone addiction, and academic satisfaction. Data were analysed using Partial Least Squares Structural Equation Modelling (PLS-SEM). Faculty support was positively associated with academic satisfaction (r = 0.334, $p < 0.001$) and negatively associated with sleep problems (r = −0.727, $p < 0.001$) and smartphone addiction (r = −0.491, $p < 0.001$). Both sleep problems and smartphone addiction were significant negative predictors of academic satisfaction. Mediation analysis showed that both variables partially mediated the relationship between faculty support and academic satisfaction. Faculty support enhances academic satisfaction not only directly but also indirectly by reducing sleep-related issues and smartphone dependency. Strengthening faculty-student relationships could serve as a strategic intervention to improve academic well-being and behavioural regulation in nursing education, particularly within low-resource settings.

**Data availability statement:** Data for this study can be found at https://doi.org/10.17605/OSF.IO/6T3X7.

**Funding:** The authors received no specific funding for this work.

**Competing interests:** The authors have declared that no competing interests exist.

## Introduction

In an era where student wellbeing and academic performance are central to higher education outcomes [1], faculty support has become increasingly recognised as a strong basis for academic satisfaction and student success [2]. Faculty support, including quality mentoring, constructive feedback, availability, and emotional encouragement, has been consistently associated with improved academic engagement [3] and positive educational experiences [4,5]. However, the learning environment of today's college students is marked by growing challenges, including digital overuse and sleep-related disturbances [6]. Globally, studies have shown that excessive smartphone usage among students is linked with impaired sleep quality [6,7], increased anxiety [8], and decreased academic performance [7,9].

These challenges are equally salient in sub-Saharan Africa, particularly in Ghana, where smartphone ownership and social media use have surged among tertiary students. Smartphone ownership in Ghana has grown rapidly, with about 34–35% of adults owning smartphones and over 80% owning mobile phones [10,11]. Among tertiary students, mobile phone ownership is much higher. For example, studies have shown that nearly all tertiary students own mobile phones, with smartphones being the dominant device used (over 85%), particularly for social media and entertainment [12,13]. Nursing students, who experience high academic and clinical demands, are especially vulnerable to the consequences of poor sleep, digital distractions and academic performance [14,15]. Despite growing attention to these issues in other regions, the Ghanaian literature remains limited, particularly concerning how institutional factors such as faculty support may influence the adverse effects of smartphone use and sleep problems. The specific interplay between these variables in nursing education, where academic satisfaction is crucial for retention and professional readiness, remains an under-explored area.

Globally, research tends to examine direct, linear relationships such as smartphone addiction's impact on academic performance or sleep problems as isolated predictors of poor outcomes [6,9,16]. Few studies have integrated these constructs into a comprehensive model that includes institutional support structures, such as faculty involvement, as potential protective factors. Furthermore, synthesised evidence shows that most existing research emanates from Western or Asian contexts, failing to account for the socio-cultural and infrastructural dynamics specific to African nursing education systems [15]. Ghanaian students face distinct stressors, including limited institutional resources, large class sizes, and often insufficient academic guidance [17]. However, research has not holistically examined how faculty support might directly or indirectly influence academic satisfaction through its impact on sleep quality and smartphone use habits.

This study therefore seeks to model the associations between faculty support and academic satisfaction among college student nurses in Ghana, focusing on the mediating roles of sleep problems and smartphone addiction. Aside addressing the knowledge gap in this area of research, this study will generate context-specific insights that can inform the development of more supportive educational environments. Knowledge on how faculty support influences academic satisfaction can guide

interventions in nursing education. Thus, in the context of global health system challenges and an ongoing need for skilled nurses, the findings from this research will contribute to both local educational reform and broader discussions on student well-being and academic success in low-resource settings.

## Theoretical framework

The most suitable theoretical framework for this study is the Social Support Theory, originally advanced by Cobb [18]. Cobb [18] argued that perceived support from others in the form of emotional, informational, or instrumental can buffer individuals against the adverse effects of stress and improve overall well-being. In academic contexts, this theory has been widely applied to examine how supportive interpersonal relationships, especially from faculty, enhance students' academic well-being [19]. In this study, faculty support reflects a form of social support that can improve sleep and reduce problematic smartphone use. Specifically, the theory underpins the hypothesis that student nurses who perceive strong faculty support are more likely to experience better academic satisfaction, both directly and indirectly, by reducing stress-induced behaviours like excessive smartphone use and promoting healthier routines like improved sleep hygiene. This framework does not only contextualises the direct effect of faculty support on student well-being but also aligns with the proposed mediated relationships in the model.

## Hypotheses development

Faculty support is a key element of the academic social environment and plays a vital role in promoting student adjustment, coping, and academic success, particularly in demanding disciplines such as nursing [20]. Emotional and informational support from authority figures helps individuals manage stress more effectively [18]. In academic settings, supportive faculty interactions enhance students' psychological resilience and capacity to cope with academic stressors [21]. Nursing students, who often face high academic workloads and clinical responsibilities, are especially susceptible to stress-induced problems such as poor sleep and digital dependency [14,15]. In the Ghanaian, where resource constraints and large class sizes are common in tertiary institutions, faculty support is particularly essential. Amponsah et al. [22] found that Ghanaian tertiary students who perceived high social support from lecturers reported significantly lower mental health burdens, suggesting a protective role of support systems in promoting student well-being.

Beyond direct academic outcomes, faculty support may influence student health-related behaviours such as sleep quality and technology use [23]. Poor sleep is a frequent complaint among university students, especially those under academic pressure [14,15]. Research indicates that inadequate sleep negatively affect memory, concentration, and academic motivation [24]. When faculty demonstrate flexibility, clarity, and emotional support, they help reduce students' cognitive overload, potentially promoting better sleep hygiene [14]. Concurrently, smartphone addiction is an emerging concern among university students globally. It is often a maladaptive coping mechanism for academic stress, loneliness, or disengagement [25]. In the Ghanaian context, evidence suggests that academic stress and low engagement significantly predicted smartphone dependency among young people [26–28]. This reinforces the idea that faculty support could indirectly reduce such tendencies by enhancing students' emotional and academic regulation.

Additionally, a substantial body of research supports the positive association between faculty support and academic satisfaction [14,15]. Faculty who provide constructive feedback, maintain accessibility, and show genuine concern for student progress tend to foster a stronger sense of belonging [29,30], well-being [31], and academic performance [32] among learners. These attributes are critical to students' satisfaction with their academic experience. Synthesised evidence shows that perceived faculty support is a strong predictor of academic satisfaction and overall student wellbeing. In the Ghanaian context, Donkor et al. [17] highlighted how instructor responsiveness and guidance significantly shaped the learning experiences and satisfaction of students, affirming the relevance of faculty support in resource-constrained educational environments. Based on this evidence, the authors argue that:

H$_1$: Faculty support is negatively associated with sleep problems.

H$_2$: Faculty support is negatively associated with smartphone addiction.

H$_3$: Faculty support is positively associated with satisfaction with academics.

Sleep disturbances impair cognitive function, emotional regulation, and academic performance, all of which directly affect satisfaction with academics [24]. Research indicates that insufficient or poor-quality sleep correlates with reduced concentration, increased irritability, and lower academic self-efficacy [24,33]. Nursing students are particularly susceptible due to night shifts during clinical placements and demanding coursework [34]. In Ghana evidence shows that poor sleep quality among tertiary students was significantly associated with lower daytime function, academic performance and motivation [35,36]. Thus, students experiencing sleep problems are less likely to feel competent and satisfied with their academic progress. Hence, authors hypothesised that:

H$_4$: Sleep problems are negatively associated with satisfaction with academics.

Smartphone addiction has been widely recognised as a disruptor of academic focus and performance [37]. It diminishes time allocated to academic tasks, increases procrastination, and correlates with increased academic stress [38] In a study by Samaha and Hawi [25], smartphone overuse among university students significantly predicted low academic satisfaction and performance, a trend echoed by more recent findings among tertiary students in Ghana [35,36]. Among student nurses where digital distractions are rising rapidly, excessive smartphone use contributes to disengagement from coursework and reduced satisfaction with academic life. Authors argued hypothesised that:

H$_5$: Smartphone addiction is negatively associated with satisfaction with academics.

Emerging research suggests a bidirectional relationship between sleep disturbances and smartphone addiction, though evidence leans toward problematic use disrupting circadian rhythms [39]. Excessive nighttime screen exposure delays melatonin production, prolongs sleep latency, and reduces sleep duration [40,41]. In a meta-analysis by Liu et al. [42], smartphone addiction was significantly associated with poor sleep quality among university students. Additionally, in Ghana, Akowuah et al. [43] found that students who reported compulsive smartphone use also reported significantly lower sleep quality. Based on this evidence, authors hypothesised that:

H$_6$: Sleep problems are positively associated with smartphone addiction.

Sleep problems are increasingly recognised as a psychological and behavioural pathway through which environmental and interpersonal factors influence academic outcomes [24]. Faculty support may alleviate psychological distress and reduce behavioural risk factors, including poor sleep hygiene, which subsequently enhances student nurses' academic experiences [44,45]. When students feel academically supported, they experience lower stress and cognitive load, which positively affects sleep quality [44–46]. Poor sleep, on the other hand, has been shown to impair memory consolidation, reduce attentional capacity, and diminish learning motivation [47,48], all of which are linked to academic dissatisfaction. Thus, the influence of faculty support on satisfaction may not be purely direct but rather transmitted through its positive effect on sleep regulation. In the Ghanaian context, nursing students usually juggle intense academic and clinical demands, so the ability of faculty to ease students' academic burden may lead to better sleep patterns, which may in turn foster higher satisfaction with academics. Hence, authors argue that:

H$_7$: Sleep problems significantly mediate the association between faculty support and academic satisfaction.

Smartphone addiction represents a maladaptive coping strategy that may explain the pathway through which limited faculty support diminishes academic satisfaction. Students lacking guidance and interpersonal connection with faculty may resort to excessive smartphone use to cope with academic stress, loneliness, or disengagement [25,49]. Faculty support, as a form of social and institutional backing, can help buffer this stress and promote purposeful academic behaviour [50]. Empirical evidence suggests that when students feel supported by lecturers and supervisors, they are less likely to develop problematic smartphone behaviours, which are known to negatively affect concentration, academic involvement, and time management [51–53]. In a mediated framework, faculty support would therefore reduce smartphone addiction, which in turn would improve academic satisfaction by enhancing academic focus and reducing distraction. This indirect

pathway is particularly relevant in the Ghanaian context, where mobile phone use is pervasive among tertiary students, but institutional support systems remain underdeveloped. Thus, testing this mediating mechanism allows stakeholders to recognise digital addiction not only as an isolated problem but as a symptom of deeper institutional gaps in student-faculty engagement. Hence, the authors hypothesised that:

H₈: Smartphone addiction significantly mediates the association between faculty support and academic satisfaction.

## Methods

### Ethics statement

The study was conducted in accordance with the ethical principles outlined in the Declaration of Helsinki and received approval from the Institutional Review Board (IRB) of the University of Cape Coast with an ethical clearance (UCCIRB/EXT/2023/36). All participants were provided with detailed information regarding the purpose, procedures, potential risks, and benefits of the study, and informed consent was obtained before data collection. Confidentiality, anonymity, and voluntary participation, including the right to withdraw at any time without penalty, were strictly upheld.

### Study design and population

This study employed a cross-sectional survey design. The target population included all regular Level 200–400 undergraduate students enrolled at the School of Nursing and Midwifery (SoNM), University of Cape Coast (UCC). Level 100 students were excluded from the study as they had not yet completed a full academic year and therefore lacked the minimum cumulative grade point average required for inclusion. Furthermore, Level 100 students are typically in an academic and social transition phase and have not fully encountered the demanding components of the nursing programme, such as clinical placements and licensure examination preparation, which were central to the stress-related constructs explored in this research. The total number of eligible students from Level 200 to Level 400 was 602.

### Study setting

The research was conducted at the SoNM, located within the UCC in the Central Region of Ghana. UCC is a prestigious public university situated along the Atlantic coastline. Since its establishment in 1962, the university has grown into a renowned centre for excellence in teacher education, research, and professional development. It offers a wide array of undergraduate and postgraduate programmes through its various faculties and institutes. Notably, in 2024, UCC was ranked the top university in both Ghana and West Africa by Times Higher Education. The SoNM, part of the College of Health and Allied Sciences (CoHAS), comprises four departments: Adult Health, Mental Health, Public Health, and Maternal and Child Health.

### Measures

Nursing students' addiction to smartphones was measured using the smartphone addiction scale [54]. An example of an item in the scale is "*I am always thinking that I have a message on my smartphone*", and it is rated on a 5-point Likert scale (1=strongly disagree, 5=strongly agree). The smartphone addiction scale is internally consistent with a Cronbach's alpha of 0.91 [54]. Among the current sample, Cronbach's alpha of 0.89 was found. Higher scores represented high smartphone addiction [55]. In addition, students' socio-demographic information, including age, level in college, sex and marital status, was collected.

Student nurses' satisfaction with academic was measured using the satisfaction with academics Scale (SAS) [56]. SAS is a 6-item subscale of the College Student Subjective Wellbeing Questionnaire (CSSWQ), developed and validated by Renshaw and Bolognino [56] to assess students' satisfaction with their academic experience. Using a 7-point Likert scale (1=strongly disagree to 7=strongly agree), the scale captures students' perceptions of their coursework, major, and academic progress. A sample item is, "*I am happy with my academic major.*" The SAS has demonstrated strong internal consistency, with Cronbach's alpha values of.88 and.92 across two validation samples [56]. Higher scores indicate greater academic satisfaction.

The *Perceived Faculty Support* subscale is one of four components of the 26-item Revised Sense of Belonging Scale, developed by Hoffman et al. [57] to assess college students' sense of belonging. This subscale contains 10 items that measure students' comfort and perceived support in their interactions with faculty. Items are rated on a 5-point Likert scale (1 = *completely untrue* to 5 = *completely true*). A sample item is: *"I feel comfortable seeking help from a teacher before or after class."* The subscale has demonstrated strong internal consistency, with a Cronbach's alpha of .89 [57]. Higher scores indicate stronger perceived faculty support, an important element in fostering student engagement, retention, and psychosocial wellbeing.

Sleep problems were assessed using the revised version of the medical outcomes study sleep scale (MOS Sleep-R) [58]. It is used to assess sleep quality and disturbances over a recall period of either one or four weeks. In this study, six items were adopted from the Sleep Problem Index I (SPI-I) of the MOS Sleep-R to measure sleep problems among college students. These items assess issues such as difficulty falling asleep, waking during the night, feeling unrested upon waking, daytime sleepiness, and not getting enough sleep. Responses are rated on a 5-point Likert scale ranging from *"All of the time"* to *"None of the time."* The MOS Sleep-R has demonstrated strong internal consistency, with Cronbach's alpha values of 0.80 [58], and is widely used for evaluating sleep-related functioning in both general and clinical populations. Higher scores indicate fewer sleep problems. The nursing students were also asked of their age, sex, marital status, level in college and religion.

## Data collection procedure

This study adopted a census approach, targeting all Level 200–400 nursing students at the School of Nursing and Midwifery, University of Cape Coast. To ensure formal access and institutional cooperation, permission was sought through the Dean of the School, the Faculty Officer, and the various Heads of Department. This approach helped secure administrative support, aligned with academic protocols, and encouraged participation by fostering trust and legitimacy among the students. Data were collected using an online questionnaire, which was distributed via students' WhatsApp platforms, a widely used and accessible medium for communication among university students. To enhance participation and provide support, three trained research assistants were recruited. Their role was to assist students who encountered challenges with accessing or completing the online forms, including those with limited digital literacy or internet access issues.

The purpose of the study was clearly explained to all prospective participants in simple, jargon-free language using a brief introductory message embedded within the online form. Informed consent was obtained electronically before participants could proceed to the questionnaire. Participation was entirely voluntary, and students were assured that they could opt out at any stage without consequence. No item was made compulsory, and they could skip items they were not comfortable answering. To ensure anonymity and confidentiality, no personal identifiers were collected. Privacy and data protection standards were upheld in line with the 1964 Helsinki Declaration and its later amendments. Data collection took place over two months, from February 20 to April 19, 2024, and achieved an impressive response rate of 93.5%, reflecting strong engagement from the student body.

## Analytical procedures

Before analysis, the dataset was screened for completeness, and missing values were assessed and addressed appropriately to ensure data integrity. For quantitative variables, missing data were handled within SmartPLS using mean replacement, which is the default method for handling missing values in continuous indicators. This approach helped retain cases while minimising potential distortion of the data structure. For categorical variables, the median of nearby points method was applied. Socio-demographic characteristics of participants were analysed using descriptive statistics, specifically mean, standard deviation, frequencies and percentages. To examine the hypothesised relationships among the study variables, Partial Least Squares Structural Equation Modelling (PLS-SEM) was employed. The analysis was conducted using SmartPLS statistical software (version 4.1.1.4), following the recommended procedures outlined by Hair et al. [59] for

PLOS Mental Health

evaluating path models. PLS-SEM was chosen for its suitability in analysing complex models with mediating and moderating effects, especially under non-normal data conditions and smaller sample sizes. Its predictive focus and flexibility made it appropriate for this exploratory study in a relatively under-researched context. The procedures for evaluating the path model include model specification, assessment of the outer and inner model, and evaluation of hypothesised paths.

## Model specification

All constructs in the path model were reflectively modelled, with each latent variable measured by multiple observed indicators. Faculty Support was treated as an exogenous construct. Sleep Problems and Smartphone Addiction served as mediating variables. Satisfaction with Academics was the endogenous outcome. The model included direct paths from Faculty Support to all other constructs, as well as indirect effects via the two mediators. Additionally, sequential mediation was modelled through the pathway Faculty Support→Sleep Problems→Smartphone Addiction→Satisfaction with Academics. See Fig 1 for the structural model.

## Assessment of the outer model

The outer model was assessed focusing on indicator reliability, internal consistency reliability, convergent validity, and discriminant validity. Items with outer loadings less than 0.70 [59] were removed to enhance measurement quality, as such items are considered to contribute weakly to their respective constructs. For example, item SAS_2 had outer loading less than.70 and hence, was deleted. After this refinement, all retained indicators had loadings above the recommended threshold of 0.70, confirming satisfactory indicator reliability. Internal consistency reliability was established, with Cronbach's alpha (α) values ranging from 0.866 to 0.943 and composite reliability (CR) values from 0.933 to 0.944 both exceeding the minimum acceptable value of 0.70 [59]. Convergent validity was also established, as the average variance extracted (AVE) for all constructs exceeded 0.50 [59], indicating that each construct explained more than half of the variance in its indicators. See details of CR, alpha and AVEs in Table 1. Discriminant validity was confirmed through the Heterotrait-Monotrait (HTMT) ratio, with all values below 1, and the Fornell-Larcker criterion, where the square root of

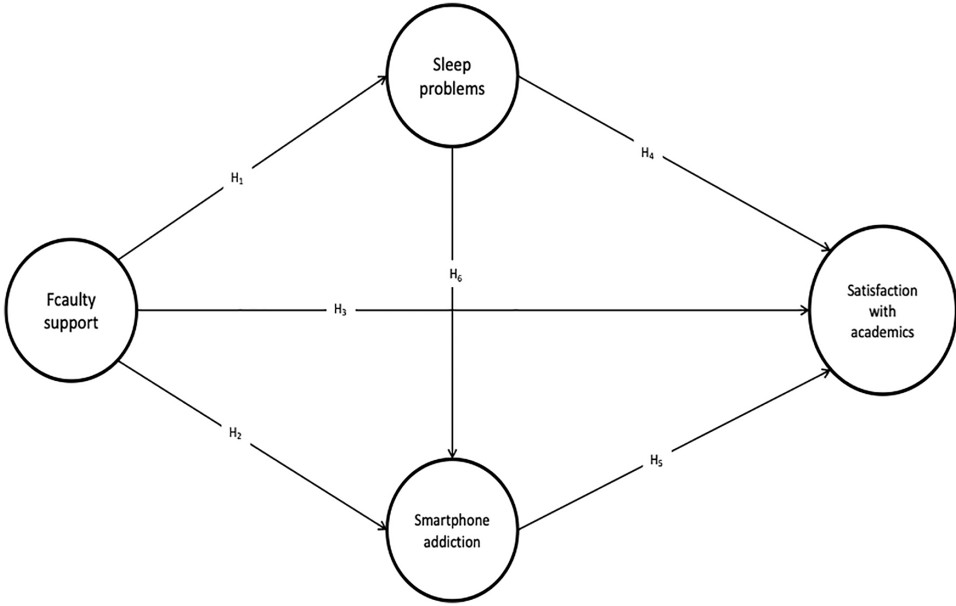

**Fig1. Structural model.**

**Table 1. Outer loading, composite reliability (CR), Cronbach's alpha (α) and average variance extracted (AVE) of constructs.**

| Constructs | Outer loading | | | |
|---|---|---|---|---|
| | Faculty support | Academic satisfaction | Sleep problem | Smartphone addiction |
| Faculty support (CR = .944, α = .943, AVE = .662) | | | | |
| FS_1 | 0.823 | | | |
| FS_2 | 0.805 | | | |
| FS_3 | 0.870 | | | |
| FS_4 | 0.782 | | | |
| FS_5 | 0.800 | | | |
| FS_6 | 0.865 | | | |
| FS_7 | 0.793 | | | |
| FS_8 | 0.854 | | | |
| FS_9 | 0.758 | | | |
| FS_10 | 0.779 | | | |
| Academic satisfaction (CR = .944, α = .866, AVE = .652) | | | | |
| SAS_1 | | 0.859 | | |
| SAS_3 | | 0.828 | | |
| SAS_4 | | 0.806 | | |
| SAS_5 | | 0.787 | | |
| SAS_6 | | 0.754 | | |
| Sleep problem (CR = .933, α = .913, AVE = .701) | | | | |
| SP_1 | | | 0.871 | |
| SP_2 | | | 0.737 | |
| SP_3 | | | 0.912 | |
| SP_4 | | | 0.814 | |
| SP_5 | | | 0.748 | |
| SP_6 | | | 0.922 | |
| Smartphone addiction (CR = .935, α = .916, AVE = .705) | | | | |
| SAS-SV_1 | | | | 0.859 |
| SAS-SV_3 | | | | 0.853 |
| SAS-SV_4 | | | | 0.815 |
| SAS-SV_5 | | | | 0.851 |
| SAS-SV_7 | | | | 0.795 |
| SAS-SV_8 | | | | 0.860 |

each construct's AVE was greater than its correlations with other constructs. This means that the measurement model is reliable and valid.

## Assessment of inner model

The inner model was assessed using key criteria, including multicollinearity, model fit, predictive relevance, model accuracy, and explanatory power (adjusted $R^2$). All Variance Inflation Factor (VIF) values ranged between 1.000 and 4.941, which are below the commonly accepted threshold of 5, indicating that multicollinearity issues were not a in the path model. The Standardised Root Mean Square Residual (SRMR) value for both the saturated and estimated models was 0.084, which is within the acceptable limit of 0.10 [59], indicating a good overall model fit. Furthermore, Stone-Geisser $Q^2$

values for the three endogenous constructs, Sleep Problems (0.525), Smartphone Addiction (0.692), and Satisfaction with Academics (0.544), were all above zero, confirming the strong predictive relevance of the model. This was done using the PLS-Predict procedure in the SmartPLS.

The predictive accuracy of the model was evaluated using the Root Mean Square Error (RMSE) and Mean Absolute Error (MAE). RMSE estimates the average magnitude of prediction errors, giving higher weight to larger errors, while MAE reflects the average size of the absolute differences between predicted and actual values. RMSE values ranged from 0.557 to 0.691, and MAE values ranged from 0.431 to 0.481 across the three outcomes, indicating acceptable levels of prediction error. The adjusted $R^2$ (co-efficient of determination) values further supported the model's explanatory power, with 52.7% of variance explained in Sleep Problems ($R^2 = 0.527$), 61.8% in Smartphone Addiction ($R^2 = 0.618$), and 79.7% in Satisfaction with Academics ($R^2 = 0.797$). These values demonstrate that the model accounted for a substantial pro-portion of variance in all three endogenous variables, confirming its robustness and adequacy for hypothesis testing and prediction.

## Assessment of path co-efficient

The model's path coefficients, significance levels, and effect sizes were assessed using the bootstrapping procedure with 5,000 resamples, following the recommendations of Hair et al. [59]. As shown in Fig 2, a path was considered sta-tistically significant if the t-value >1.96, at p value < 0.05. The strength and direction of the relationships were interpreted using the standardised path coefficients, which represent the direct effects between constructs. Additionally, the practical significance of each relationship was evaluated using Cohen's $f^2$, with values of 0.02, 0.15, and 0.35 interpreted as small, moderate, and large effect sizes, respectively [60].

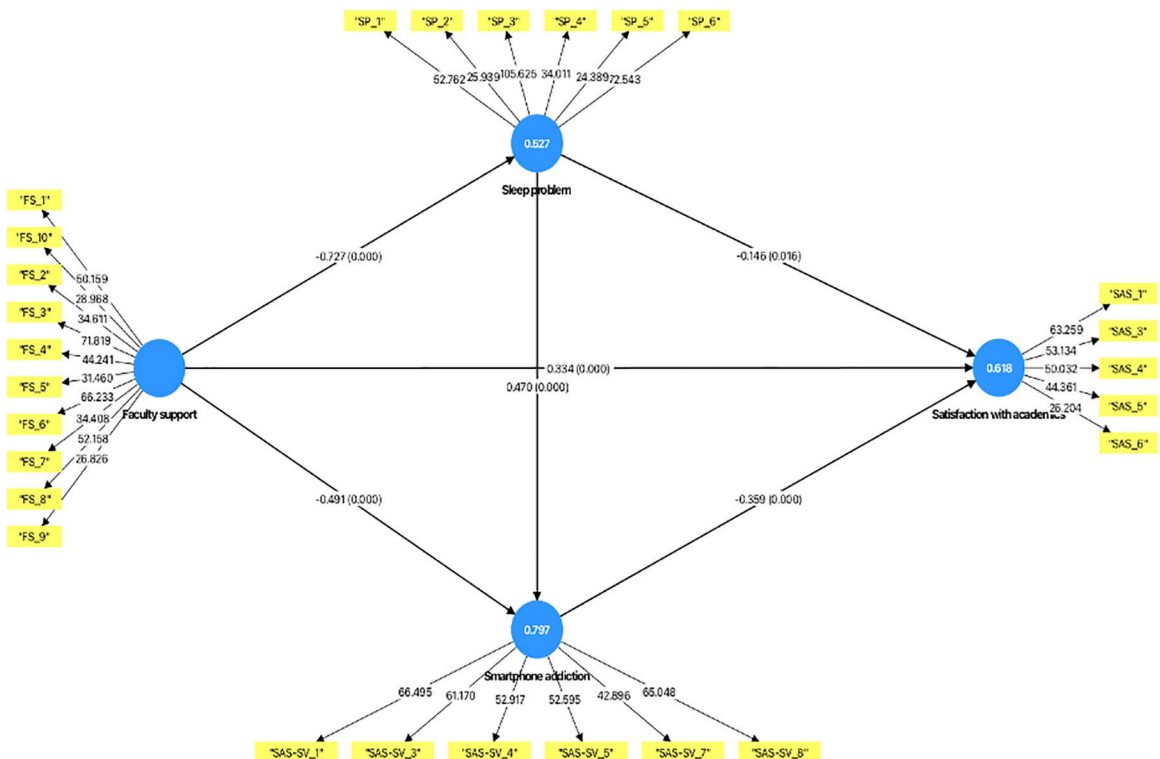

**Fig 2. Outer weights, path co-efficient and adjusted R² of path model.**

## Assessments of mediation paths

According to Hair et al. [59], the assessment of the mediation oaths begins by examining the significance of the direct paths between the independent and dependent variables. If the direct effect remained significant but was reduced after introducing the mediating variable, this indicated partial mediation. However, if the direct effect became non-significant when the mediator was included, full mediation was assumed. The presence and strength of mediation were determined by evaluating both the direct and indirect effects along with their corresponding confidence intervals and p-values.

## Results

### Socio-demographic information

Socio-demographic information of participants is presented in Table 2. From the table, the majority of the participants were female (78.5%), with 21.5% being male. The mean age of respondents was 25.5 years (SD = 3.88). Most participants were single (90.8%), while 9.2% were married. In terms of religious affiliation, the majority identified as Christians (90.9%), followed by Muslims (8.0%) and a small proportion practising traditional religion (1.1%). Regarding academic level, 35.2% of the students were in Level 300, 33.2% in Level 200, and 31.6% in Level 400.

### Hypotheses testing

Table 3 presents the summary of the six hypothesised associations (H₁–H₆). These hypotheses were statistically supported. The first hypothesis (H₁) showed a positive association between faculty support and satisfaction with academics ($r = 0.334$, $t = 5.672$, $p < 0.001$, $f^2 = 0.089$). Also, there is a negative association between faculty support and sleep problems (H₂) ($r = -0.727$, $t = 29.966$, $p < 0.001$, $f^2 = 1.118$), and H₃ showed a negative association with Smartphone Addiction ($r = -0.491$, $t = 16.568$, $p < 0.001$, $f^2 = 0.563$). Furthermore, sleep problems were negatively associated with satisfaction with academics but had an insignificant effect size (H₄) ($r = -0.146$, $t = 2.419$, $p = 0.016$, $f^2 = 0.017$). The analysis also revealed a positive association between sleep problems and smartphone addiction (H₅) ($r = 0.470$, $t = 15.571$, $p < 0.001$, $f^2 = 0.515$). Finally, the analysis revealed a negative relationship between smartphone addiction and satisfaction with academics (H₆) ($r = -0.359$, $t = 5.590$, $p < 0.001$, $f^2 = 0.069$). See Table 3 for details.

**Table 2. Socio-demographic information.**

| Variables | Categories | Frequency | Percentage (%) | Mean | SD |
|---|---|---|---|---|---|
| Gender | Male | 121 | 21.5% | | |
| | Female | 442 | 78.5% | | |
| Age | | | | 25.5 | 3.88 |
| Marital Status | Married | 52 | 9.2% | | |
| | Single | 511 | 90.8% | | |
| Religion | Christianity | 512 | 90.9% | | |
| | Islam | 45 | 8.0% | | |
| | Traditional | 6 | 1.1% | | |
| Level in college | 200 | 187 | 33.2% | | |
| | 300 | 198 | 35.2% | | |
| | 400 | 178 | 31.6% | | |

SD, Standard Deviation

**Table 3. Path coefficients and effect size with confidence intervals.**

| Paths | Path coefficients | 95%Confidence interval | | T-statistics | P-values | f² | 95%Confidence Interval | | T-statistics | P-values |
|---|---|---|---|---|---|---|---|---|---|---|
| | | Lower limit | Upper limit | | | | Lower limit | Upper limit | | |
| Faculty support -> Satisfaction with academics | 0.334 | 0.226 | 0.457 | 5.672 | 0.000 | 0.089 | 0.041 | 0.166 | 2.741 | 0.006 |
| Faculty support -> Sleep problem | -0.727 | -0.774 | -0.680 | 29.966 | 0.000 | 1.118 | 0.859 | 1.498 | 6.821 | 0.000 |
| Faculty support -> Smartphone addiction | -0.491 | -0.549 | -0.433 | 16.568 | 0.000 | 0.563 | 0.398 | 0.783 | 5.733 | 0.000 |
| Sleep problem -> Satisfaction with academics | -0.146 | -0.265 | -0.030 | 2.419 | 0.016 | 0.017 | 0.001 | 0.059 | 1.131 | 0.258 |
| Sleep problem -> Smartphone addiction | 0.470 | 0.409 | 0.527 | 15.571 | 0.000 | 0.515 | 0.350 | 0.728 | 5.306 | 0.000 |
| Smartphone addiction -> Satisfaction with academics | -0.359 | -0.478 | -0.231 | 5.590 | 0.000 | 0.069 | 0.026 | 0.131 | 2.535 | 0.011 |

## Mediation results

Table 4 presents the results of the hypothesised mediation effects. The indirect path faculty support→sleep problems→satisfaction with academics ($H_7$) was significant ($r = -0.229$, $t = 4.884$, $p < 0.001$), indicating that faculty support is associated with reduced sleep problems, which in turn relate to higher academic satisfaction. The direct path remained significant, so this reflects partial mediation. Similarly, the indirect path faculty support→smartphone addiction→satisfaction with academics ($H_8$) was also significant ($r = 0.177$, $t = 5.725$, $p < 0.001$), suggesting that reduced smartphone addiction, linked to higher faculty support, is associated with greater academic satisfaction. The persistence of the direct effect again indicates partial mediation.

## Summary of findings

The findings confirmed all six hypothesised associations, showing that faculty support was positively associated with academic satisfaction and negatively associated with sleep problems and smartphone addiction. Sleep problems and smartphone addiction were also significantly related to academic satisfaction. Mediation analysis further revealed that both sleep problems and smartphone addiction partially mediated the relationship between faculty support and academic satisfaction, highlighting their roles as important indirect pathways.

**Table 4. Results summary for hypothesized mediation paths.**

| Paths | Path coefficient | 95% confidence interval | | T statistics | P-value |
|---|---|---|---|---|---|
| | | Lower limit | Upper limit | | |
| **Total effects** | | | | | |
| Faculty support -> Satisfaction with academics | 0.740 | 0.695 | 0.784 | 32.927 | 0.000 |
| Faculty support -> Sleep problem | -0.727 | -0.774 | -0.680 | 29.966 | 0.000 |
| Faculty support -> Smartphone addiction | -0.833 | -0.861 | -0.803 | 56.934 | 0.000 |
| Sleep problem -> Satisfaction with academics | -0.314 | -0.422 | -0.205 | 5.754 | 0.000 |
| Smartphone addiction -> Satisfaction with academics | -0.359 | -0.478 | -0.231 | 5.590 | 0.000 |
| **Total indirect effect** | | | | | |
| Faculty support -> Satisfaction with academics | 0.405 | 0.310 | 0.489 | 8.925 | 0.000 |
| **Specific Indirect Effect** | | | | | |
| Faculty support -> Sleep problem -> Smartphone addiction | -0.341 | -0.385 | -0.300 | 15.566 | 0.000 |
| Faculty support -> Smartphone addiction -> Satisfaction with academics | 0.177 | 0.115 | 0.233 | 5.725 | 0.000 |

## Discussion

The findings from this study confirmed that faculty support is significantly associated with higher satisfaction with academics and lower levels of sleep problems and smartphone addiction among student nurses in Ghana. These associations align with Social Support Theory [18]. The positive association between faculty support and academic satisfaction reflects the growing body of evidence that highlights the role of supportive academic relationships in fostering positive educational experiences [2,3]. In Ghanaian schools characterised by large classes and limited resources, the value of accessible and emotionally supportive faculty is especially salient [17,28]. The findings reinforce earlier studies showing that students who feel supported by faculty are more likely to engage in their academic work, manage their academic challenges more effectively, and report greater satisfaction with their educational environment [29,30]. Moreover, the strength of the association between faculty support and reduced sleep problems and smartphone addiction indicates its broader psychosocial influence beyond purely academic outcomes. This also suggests that faculty support functions not only as an academic enabler but as a form of emotional scaffolding that promotes behavioural self-regulation [3]. Importantly, in resource-limited educational settings, faculty support may act as an informal substitute for institutional well-being services that are often underfunded or absent.

The mediating roles of sleep problems and smartphone addiction further illustrate the psychological and behavioural mechanisms through which faculty support relates to academic satisfaction. This suggests that while faculty support directly contributes to students' satisfaction, it also operates indirectly by influencing sleep and digital behaviours. In particular, the finding that reduced sleep problems are associated with higher academic satisfaction is consistent with existing evidence that inadequate sleep impairs memory, learning motivation, and emotional regulation [24,33]. Among nursing students, the protective effect of faculty support on sleep could be crucial due to rigorous class hours and clinical placements. Faculty who are flexible, responsive, and empathetic may reduce students' stress load, leading to better sleep hygiene and, consequently, improved academic well-being [14,44]. In the Ghanaian context, this is relevant as students often face academic pressures without the benefit of strong institutional welfare systems. The mediating role of sleep, therefore, illustrates how psychosocial support can translate into physiological and behavioural outcomes that ultimately affect academic experiences.

Smartphone addiction also significantly mediated the relationship between faculty support and academic satisfaction. This is essential in a setting where smartphone use among tertiary students is nearly universal [27] and often serves both academic and entertainment purposes. Excessive use, however, has been linked to academic disengagement and poor performance [7,25]. The current findings suggest that when students receive less faculty attention or experience a weak academic connection, they may be more prone to maladaptive behaviours such as compulsive smartphone use [26,49]. Faculty support, by enhancing students' emotional regulation and academic motivation, may reduce reliance on digital devices as a coping mechanism [7]. In this light, smartphone addiction becomes not merely a personal failing but a behavioural indicator of unmet academic and emotional needs. In Ghana's educational context, this highlights the systemic role of faculty in shaping not only academic but also behavioural outcomes among students. Hence, the model presented in this study suggests a more integrated understanding of how institutional support can impact student satisfaction through its influence on modifiable behavioural and psychological factors.

### Implications for nursing education and research

The implications of this study extend beyond confirming known associations and point to the urgent need for nursing education systems, especially in resource-limited contexts like Ghana, to formally integrate faculty support as a strategic pillar of academic success and student wellbeing. Rather than treating support as incidental or personality-driven, institutions should embed structured mentorship, academic advising, and psychosocial check-ins into faculty roles and training. This shifts faculty-student interaction from a transactional model to a developmental one, where guidance is continuous and

proactive. Given the known constraints such as large class sizes, limited mental health infrastructure, and digital overexposure among students, faculty support must be leveraged not only for pedagogical purposes but as a buffer against the psychosocial strain that undermines learning. Institutional investment should therefore focus on workload balancing for faculty, incentivised mentorship, and teaching evaluations that include support-related competencies. For nursing education research, these findings call for longitudinal and implementation-focused studies that examine how faculty support interventions, whether policy-driven or informal, affect outcomes such as resilience, retention, and professional readiness. Moreover, future research should unpack contextual nuances, including gender, socio-economic status, and digital behaviour patterns, that may shape how support is experienced and utilised in African nursing education systems.

## Limitation in this review

This study was cross-sectional in nature, which limits causal interpretations of the observed relationships between faculty support, sleep problems, smartphone addiction, and academic satisfaction. Self-reported measures were used for all constructs, introducing the possibility of social desirability and recall biases. Additionally, the study was conducted within a single nursing school in Ghana, which may limit the generalisability of the findings to other institutions or disciplines. While the sample had a high response rate, future studies should consider more diverse and multi-institutional samples as well as longitudinal or experimental designs to examine changes over time and test intervention effects.

## Conclusion

The findings of this study suggest a critical role of faculty support in enhancing academic satisfaction among nursing students, both directly and indirectly through improved sleep quality and reduced smartphone addiction. Faculty support emerged not only as a protective academic factor but also as a psychosocial buffer that influences student well-being. Sleep problems and smartphone addiction served as significant mediators, illustrating the behavioural and psychological mechanisms that underlie student experiences. These insights indicate the need for nursing education stakeholders, particularly in resource-limited contexts, to prioritise structured faculty-student support systems as part of academic quality and student retention strategies.

## Supporting information

**S1 Text. Supplementary file.**
(DOCX)

## Author contributions

**Conceptualization:** Jerry Paul K. Ninnoni, Ignatius N. Ijere, Isaac Tetteh Commey.

**Data curation:** Ignatius N. Ijere, Isaac Tetteh Commey.

**Formal analysis:** Mustapha Amoadu.

**Investigation:** Ignatius N. Ijere.

**Methodology:** Jerry Paul K. Ninnoni, Ignatius N. Ijere, Mustapha Amoadu.

**Project administration:** Jerry Paul K. Ninnoni.

**Resources:** Ignatius N. Ijere.

**Software:** Jerry Paul K. Ninnoni, Mustapha Amoadu.

**Writing – original draft:** Isaac Tetteh Commey, Mustapha Amoadu.

**Writing – review & editing:** Mustapha Amoadu.

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
