## [Decision Letter · Decision Letter 0]

12 Dec 2025

PMEN-D-25-00489

Modelling the associations between faculty support and academic satisfaction among nursing students: mediating roles of sleep problems and smartphone addiction

PLOS Mental Health

Dear Dr. Commey,

Thank you for submitting your manuscript to PLOS Mental Health. After careful consideration, we feel that it has merit but does not fully meet PLOS Mental Health’s publication criteria as it currently stands. Therefore, we invite you to submit a revised version of the manuscript that addresses the points raised during the review process.

We look forward to receiving your revised manuscript.

Kind regards,

Robert T Rubin, MD, PhD

Academic Editor

PLOS Mental Health

Journal Requirements:

Additional Editor Comments (if provided):

Reviewers' comments:

Reviewer's Responses to Questions

**Comments to the Author**

1. Does this manuscript meet PLOS Mental Health’s publication criteria ? Is the manuscript technically sound, and do the data support the conclusions? The manuscript must describe methodologically and ethically rigorous research with conclusions that are appropriately drawn based on the data presented.

Reviewer #1: Yes

Reviewer #2: Yes

2. Has the statistical analysis been performed appropriately and rigorously?

Reviewer #1: No

Reviewer #2: Yes

3. Have the authors made all data underlying the findings in their manuscript fully available (please refer to the Data Availability Statement at the start of the manuscript PDF file)?

Reviewer #1: No

Reviewer #2: Yes

4. Is the manuscript presented in an intelligible fashion and written in standard English?

Reviewer #1: Yes

Reviewer #2: Yes

Reviewer #1: Comments to author

Title: Modelling the associations between faculty support and academic satisfaction among nursing students: mediating roles of sleep problems and smartphone addiction.

This study is interesting; the topic is doable and relevant, and also it is a study that incorporates the time factor. However, there are many points that need more clarifications; the manuscript is also too long and contains formatting inconsistencies, repetitions, and some conceptual ambiguities. Revision is needed.

In the introduction, “this study will generate context-specific insights that can inform the development of more supportive educational environments.” Have you generated the context-specific insight yet?

Is the smartphone addiction scale is for adolescents? Please justify its use for nursing students (adults).

Statistical model concerns

You mentioned the p-value in the abstract and inside the paper; unfortunately, I did not have the opportunity to see it. Please attach it in the manuscript or in the supplementary.

An unclear situation has happened between sleep problems and smartphone addiction (positive sign) versus the previous justification. Explain it logically.

Reflective measurement model indicators were dropped solely based on loading <0.70; this may alter scale validity. Please report which items were deleted and assess content validity impact.

The interpretation of mediation is right, though the sign on the smartphone path (positive r = 0.177) negates the previous negative impact. Please verify.

Discussion is similar to the introduction in many ways.

The claims of causality are not permitted, as the design is cross-sectional. Please do not exaggerate the role of faculty support as an alternative to institutional support without empirical support.

The data availability statement is a contradiction: it provides a statement that data would be available when requested, but also it has an OSF link. Please make this consistent.

Minor comments

Some references are outdated; please use the current reference as much as possible

Explain the abbreviations on first use (e.g., SEM, PLS-SEM)

Write the correct spellings for words like: “authords” “revelaed”, “stadard”, “perforamnce”,.

Thank you so much

Reviewer #2: This is a study of the association between nursing students’ perceived faculty support and their academic satisfaction, with sleep difficulty and smartphone addiction considered as mediating variables. Perceived faculty support was significantly positively related to academic satisfaction and significantly negatively associated with sleep difficulty and smartphone addiction. It was concluded that faculty support can enhance academic satisfaction and can serve as a strategic intervention in nursing education.

General comments: This is a worthwhile study, especially for resource-limited countries such as Ghana. As it is now written, it is too verbose throughout. Eight hypotheses are not needed to test the important relationships. The important findings are in Table 3. There are spelling and grammatical errors throughout. Specific comments follow:

Introduction: The introduction is too long and repetitive. Eight hypotheses are not needed for this study – one is sufficient, as stated at the top of p. 5: “…the hypothesis that student nurses who perceive strong faculty support are more likely to experience better academic satisfaction, both directly and indirectly, by reducing stress-induced behaviours like excessive smartphone use and promoting healthier routines like improved sleep hygiene.” The statistical analysis will parse the relationships.

P. 3, last paragraph, line 5: Need a sentence break between higher and studies.

P. 4, second paragraph: “Few studies have integrated these constructs into a comprehensive model that includes institutional support structures, such as faculty involvement, as potential protective factors.” This is an important feature of the present study. Emphasize this in a more succinct introduction.

P.5, last two lines are not a complete sentence.

P.6, line 1: reinforces (singular).

P.6, paragraph 2, line 4: performance is misspelled.

P.6, paragraph 2, last line: author is misspelled.

P.6, Hypotheses 1-3 can be combined into a single hypothesis: “Faculty support is positively associated with satisfaction with academics and negatively associated with sleep problems and smartphone addiction.” The subsidiary hypotheses 4-8 are redundant and contribute little to the overall issue of the importance of faculty support to nursing students’ well-being. As well, their accompanying text can be condensed considerably in the description of the single hypothesis, as suggested above.

Methods: P. 10, paragraph 1, last sentence regarding students’ demographics is redundant; this already has been stated in the Introduction.

Analytical Procedures: Overall, the reporting is very statistically dense and will be difficult for the average reader of PLOS Mental Health to comprehend. Can this section be simplified?

P. 11, paragraph 1, first sentence: “…missing values were assessed and addressed appropriately…” How did you do this? Without the procedure, I can’t judge if they were done appropriately.

P. 12, paragraph 1, line 10: Delete “the.”

P. 12, paragraph 2, line 4: “…multicollinearity issues was not a in the path model…” Was should be were. Also, word missing after “not a” – not a what?

P. 13, paragraph 2, line 2: Need space between procedure and with.

P. 13, paragraph 3, line 1: Paths misspelled.

Results: These need to be re-written after the number of hypotheses is reduced to the single, important one.

Discussion: Discussion should be singular (not Discussions).

The discussion is of reasonable length, but it should be revised in accordance with the revised Results section.

Limitation in this review: I suggest renaming this to “Strengths and Limitations” and placing it before the section, “Implications for Nursing Education and Research.” There are strengths to this report that should be highlighted, as well as limitations. Some of the strengths have been noted in the above comments. Re-emphasize these.

Conclusion: Conclusions (should be plural).

Line 5: Should be “…mediators, illustrating two of the behavioural…” There are likely others not examined.

Declarations: These are abbreviations rather than declarations.

Author contribution: contributions (should be plural). I suggest adding at the end of this paragraph a statement about any use of artificial intelligence in the literature review and/or the writing of the article.

References: Please format the references consistently, in journal format. Some doi's seem excessively long; please check them, e.g., ref's 2 & 4.

Figures and Tables: Figure 1 (structural model) is useful.

The main elements of Figure 2 (e.g., path coefficients) can be incorporated into Figure 1, and Figure 2 can be eliminated.

Table 1 can be simplified in accordance with the reduction of 8 hypotheses to one overarching hypothesis, as described above.

Table 2 should be Table 1 and its discussion re-ordered in the text.

(An interesting, later analysis might be to examine whether any of the demographic variables moderated the findings; e.g., did male students have a different pattern than females? Did college level moderate the findings? Age, marital status, and religion do not have sufficiently broad distributions to consider them as possible moderating variables.)

Table 3 is useful as a complement to Figure 1.

Table 4 has different (hypothesized) path coefficients than Table 3 for the same relationships, so I’m not clear on which is more important. When the overall analysis is simplified according to a single, overarching hypothesis, Tables 3 and 4 can be revised.

**Do you want your identity to be public for this peer review?** For information about this choice, including consent withdrawal, please see our Privacy Policy .

Reviewer #1: **Yes:** Yoseph KassaYoseph Kassa

Reviewer #2: No

Figure Resubmissions:

---

## [Decision Letter · Decision Letter 1]

27 Jan 2026

PMEN-D-25-00489R1

Modelling the associations between faculty support and academic satisfaction among nursing students: mediating roles of sleep problems and smartphone addiction

PLOS Mental Health

Dear Dr. Commey,

Thank you for submitting your revised manuscript to PLOS Mental Health. It is considerably improved,  but it still requires minor revision. Therefore, we invite you to submit a revised version of the manuscript that addresses the points raised during the review process.

Reviewer comments to author:

In their revision, the authors have cogently defended their detailed mediation analysis that includes 8 hypotheses. Whereas I suggested a single, overarching hyoothesis for ease of statistical analysis and understanding by readership, the authors have convincingly rebutted my suggestion. To their credit, the abstract is readily understandable by statistically non-sophisticated readers.

On the other hand, I commented on many details that were discrepant, including multiple spelling errors, erroneous DOIs in the references, etc. While most have been corrected, there still are some remaining. For example, the DOI for ref. 29, suspiciously long, does not work, at least for me. Every DOI in the reference list needs to be checked again, because each one must work.

The authors also indicate they provided a statement about the use of AI after the listing of author contributions, but it is missing from the revised manuscript. Please provide a detailed statement regarding whether AI was used at any stage of manuscript preparation, including the author responses to the reviews of the original version.

We look forward to receiving your revised manuscript.

With kind regards,

Robert T Rubin, MD, PhD

Academic Editor

PLOS Mental Health

Journal Requirements:

Additional Editor Comments (if provided):  N/A

Reviewers' comments:  See above.

Reviewer's Responses to Questions

**Comments to the Author**

Reviewer #1: All comments have been addressed

Reviewer #2: (See above.)

publication criteria ? Is the manuscript technically sound, and do the data support the conclusions? The manuscript must describe methodologically and ethically rigorous research with conclusions that are appropriately drawn based on the data presented.

Reviewer #1: Yes

Reviewer #2: Yes

3. Has the statistical analysis been performed appropriately and rigorously?

Reviewer #1: Yes

Reviewer #2: Yes

4. Have the authors made all data underlying the findings in their manuscript fully available (please refer to the Data Availability Statement at the start of the manuscript PDF file)?

Reviewer #1: Yes

Reviewer #2: Yes

5. Is the manuscript presented in an intelligible fashion and written in standard English?

Reviewer #1: Yes

Reviewer #2: Yes

Reviewer #1: My comments are already addressed

Reviewer #2: In their revision, the authors have cogently defended their detailed mediation analysis that includes 8 hypotheses. Whereas I suggested a single, overarching hyoothesis for ease of statistical analysis and understanding by readership, the authors have convincingly rebutted my argument. To their credit, the abstract is readily understandable by statistically non-sophisticated readers.

On the other hand, I commented on many details that were discrepant, including multiple spelling errors, erroneous DOIs in the references, etc. While most have been corrected, there still are some remaining. For example, the DOI for ref. 29, suspiciously long, does not work, at least for me. Every DOI in the reference list needs to be checked again, because each one must work.

The authors also indicate they provided a statement about the use of AI after the listing of author contributions, but it is missing from the revised manuscript. Please provide a detailed statement regarding whether AI was used at any stage of manuscript preparation, including the author responses to the reviews of the original version.

**Do you want your identity to be public for this peer review?** For information about this choice, including consent withdrawal, please see our Privacy Policy .

Reviewer #1: **Yes:** Yoseph kassaYoseph kassa

Reviewer #2: No

Figure Resubmissions:

---

## [Editor Report · Decision Letter 2]

4 Feb 2026

PMEN-D-25-00489R2

Modelling the associations between faculty support and academic satisfaction among nursing students: mediating roles of sleep problems and smartphone addiction

PLOS Mental Health

Dear Mr. Commey,

Thank you for submitting your second revisioon to PLOS Mental Health. After careful consideration, we feel that it has merit but does not fully meet PLOS Mental Health’s publication criteria as it currently stands. Therefore, we invite you to submit a revised version of the manuscript that addresses the points raised during the review process.

The latest submission is 131 pages and contains several copies of the manuscript, the last of which is a markup copy indicating the revisions of the DOIs that reviewer 2 indicated were necessary.  Regarding the use of AI, its use was explained in the responses to reviewer 2, but not presented in the MS, as the authors had said was done in a previous version, after the listing of author contributions.  Please add the AI statement to the manuscript and submit clearly marked final versions, both markup and clean.  You can remove earlier versions from your next submission to avoid ambiguity.  And provide a statement that the AI information was added.  Thank you.

A statement regarding the addition of AI information. You should upload this as a separate file labeled 'Response to Reviewers'.A marked-up copy of your manuscript that highlights any new changes. You should upload this as a separate file labeled 'Revised Manuscript with Track Changes'.An unmarked version of the same revision without tracked changes. You should upload this as a separate file labeled 'Manuscript'.

We look forward to receiving your revised manuscript.

Kind regards,

Robert T Rubin, MD, PhD

Academic Editor

PLOS Mental Health
---

## [Editor Report · Decision Letter 3]

12 Feb 2026

PMEN-D-25-00489R3

Modelling the associations between faculty support and academic satisfaction among nursing students: mediating roles of sleep problems and smartphone addiction

PLOS Mental Health

Dear Mr. Commey,

Thank you for submitting your manuscript to PLOS Mental Health. After careful consideration, we feel that it has merit but still does not fully meet PLOS Mental Health’s publication criteria. Therefore, we ask you to submit a carefully revised version that addresses the remaining points raised during the review process, as indicated below:

It is frustrating to read revision 3, because it does not address several issues pointed out in earlier revisions:

The cover letter apparently is the original version, not dated, and does not address the issue of inclusion of an A.I. statement.

"Discussions" should be singular.

"Declarations" should be "Abbreviations" (the authors indicated they had made this change, but apparently not).

There still is no statement about the use of A.I. in the manuscript. Please put it in, as requested.

Let's get the manuscript finalized, please, so we can move forward. Thank you.

A letter that responds to each point raised by the editor and reviewer(s). You should upload this letter as a separate file labeled 'Response to Reviewers'.A marked-up copy of your manuscript that highlights changes made to the original version. You should upload this as a separate file labeled 'Revised Manuscript with Track Changes'.An unmarked version of your revised paper without tracked changes. You should upload this as a separate file labeled 'Manuscript'.Please include only the above, not earlier versions of the manuscript.

We look forward to receiving your revised manuscript.

Kind regards,

Robert T Rubin, MD, PhD

Academic Editor

PLOS Mental Health

Journal Requirements:

Additional Editor Comments (if provided):  Please see comments above.
---

## [Editor Report · Decision Letter 4]

3 Mar 2026

PMEN-D-25-00489R4

Modelling the associations between faculty support and academic satisfaction among nursing students: mediating roles of sleep problems and smartphone addiction

PLOS Mental Health

Dear Mr. Commey,

It is quite frustrating to receive a deficient 4th revision of your manuscript, in which you were asked to address three minor comments:  1) Make the heading "Discussions" singular.  2) Change "Declarations" to "Abbreviations."  3) Add your statement about the use of A.I.

Only the statement about the use of A.I. is now included, and it is in the wrong place in the manuscript, having been added after the references.  It belongs in the sections after the Conclusion paragraph, not at the end of the references.  The two simple changes requested in 1) and 2) above have not been made.

As your Academic Editor, I must say this is very poor tradecraft.  Do you not want your manuscript to be as scholarly as possible?  Why are you not attending to these simple changes?

From my standpoint, I believe I have carried this manuscript forward as much as possible.  I now leave it to the journal administrative and executive editors to decide if the changes requested in 1) and 2), and moving the A.I. statement to its proper place in the manuscript, can be made in the editorial office.  If so, the manuscript is acceptable for publication.  if not, a different editorial decision must be made.

Please see comments above.

If the journal editorial office requests another revision, they will so advise me.

Yours sincerely,

Robert T Rubin, MD, PhD

Academic Editor

PLOS Mental Health

Journal Requirements:

Additional Editor Comments (if provided):

**I ask the journal editorial office to review the above and make a recommendation about further processing.**

Reviewers' comments:

Not applicable.

---

## [Editor Report · Decision Letter 5]

12 Mar 2026

Modelling the associations between faculty support and academic satisfaction among nursing students: mediating roles of sleep problems and smartphone addiction

PMEN-D-25-00489R5

Dear Mr. Commey,

We are pleased to inform you that your manuscript 'Modelling the associations between faculty support and academic satisfaction among nursing students: mediating roles of sleep problems and smartphone addiction' has been provisionally accepted for publication in PLOS Mental Health.

Best regards,

Robert T Rubin, MD, PhD

Academic Editor

PLOS Mental Health